# Temporal Grounding as a Learning Signal for Referring Video Object Segmentation

## Abstract

Referring Video Object Segmentation (RVOS) aims to segment and track objects in videos based on natural language expressions, requiring precise alignment between visual content and textual queries. However, existing methods often suffer from semantic misalignment, largely due to indiscriminate frame sampling and supervision of all visible objects during training—regardless of their actual relevance to the expression. We identify the core problem as the absence of an explicit **temporal learning signal** in conventional training paradigms. To address this, we introduce **MeViS-M**, a dataset built upon the challenging MeViS benchmark, where we manually annotate temporal spans when each object is referred to by the expression. These annotations provide a direct, semantically grounded supervision signal that was previously missing. To leverage this signal, we propose **Temporally Grounded Learning (TGL)**, a novel learning framework that directly incorporates temporal grounding into the training process. Within this framework, we introduce two key strategies. First, **Moment-guided Dual-path Propagation (MDP)** improves both grounding and tracking by decoupling language-guided segmentation for relevant moments from language-agnostic propagation for others. Second, **Object-level Selective Supervision (OSS)** supervises only the objects temporally aligned with the expression in each training clip, thereby reducing semantic noise and reinforcing language-conditioned learning. Extensive experiments demonstrate that our TGL framework effectively leverages temporal signal to establish a new state-of-the-art on the challenging MeViS benchmark. We will make our code and the MeViS-M dataset publicly available.

## 1 Introduction

Referring Video Object Segmentation (RVOS) is a challenging task focused on identifying and segmenting objects in a video sequence based on a given language description. This task has earned significant attention due to its wide-ranging applications in areas such as video editing and human-robot interaction. Unlike conventional video segmentation tasks, such as Video Instance Segmentation (VIS) (Yang et al., 2019) and Video Object Segmentation (VOS) (Perazzi et al., 2016), RVOS requires a sophisticated cross-modal understanding to accurately localize and track target objects guided by linguistic descriptions. Recent works (Ding et al., 2023; He & Ding, 2024) highlight the inherent challenges of RVOS, especially in dynamically modeling object trajectories to match nuanced language descriptions and complex motion patterns.

With the growing interest in aligning visual and linguistic modalities, recent studies have made significant strides by integrating text prompts with transformer-based architectures (Wu et al., 2022; Botach et al., 2022) and leveraging the powerful generalization capabilities of foundation models like SAM and SAM2 (Yan et al., 2024; Cuttano et al., 2025; Gong et al., 2025; Lin et al., 2025). These advancements have led to impressive performance on established benchmarks such as Ref-YouTube-VOS (Seo et al., 2020) and Ref-DAVIS (Khoreva et al., 2019), demonstrating a strong ability to segment objects based on their appearance and context. However, this progress has not translated to more complex benchmarks like MeViS (Ding et al., 2023), which features numerous similar objects and relies on motion-centric expressions for disambiguation. The performance gap on MeViS exposes a fundamental flaw in the conventional training paradigm: the reliance on indiscriminate frame sampling. As illustrated in Figure 1-(a), this approach forces a model to learn from an object's ground-truth mask even in frames where the object is not performing the action

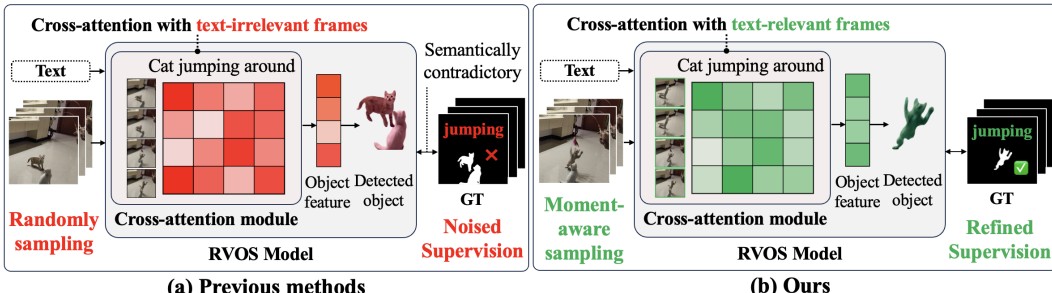

Figure 1: **Importance of moment-aware approach.** (a) Most existing RVOS methods rely on random frame sampling during training, leading to unnatural learning dynamics by forcing models to segment referred objects even in frames unrelated to the given text. (b) Our method explicitly focuses on text-relevant moments to enable semantically and temporally grounded segmentation.

described in the text. For instance, given the expression "a cat jumping around," the model is often supervised with the target cat's mask on frames where it is merely sitting still. This creates a **semantically contradictory learning signal**, teaching the model spurious correlations between the concept of "jumping" and the visual appearance of a "sitting cat." Such a flawed learning objective fundamentally prevents the model from developing a genuine understanding of the video content.

We argue that rectifying this flawed objective requires a paradigm shift: treating temporal grounding not as an afterthought, but as a primary **learning signal**. To this end, we introduce **MeViS-M**, a new dataset that augments MeViS with manually annotated frame intervals specifying the temporal scope of each expression. These annotations provide explicit supervision signals for aligning language with the most relevant video segments. Building on this dataset, we propose **Temporally Grounded Learning (TGL)**, a novel learning framework designed to effectively leverage this new signal. As illustrated in Figure 1-(b), TGL directly incorporates temporal grounding into the training process to foster a genuine understanding of video content.

Our TGL framework is composed of two key strategies that work in synergy. First, **Moment-guided Dual-path Propagation (MDP)** improves both grounding and tracking by decoupling the learning process: it applies language-guided segmentation for text-relevant moments while using language-agnostic propagation for all other moments. This ensures that linguistic features are only applied when semantically appropriate. Second, **Object-level Selective Supervision (OSS)** refines the supervision signal itself by ensuring the model only learns from objects that are temporally aligned with the expression within a given clip. This directly prevents the model from learning the spurious correlations discussed earlier. Through these strategies, our framework achieves state-of-the-art performance on the challenging MeViS benchmark. Our main contributions are summarized as follows:

1. We identify the absence of a temporal learning signal as a core problem in RVOS and introduce **MeViS-M**, a new dataset to provide explicit, object-level temporal annotations on the challenging MeViS benchmark.

2. We propose the **Temporally Grounded Learning (TGL)**, a new learning paradigm that directly leverages temporal grounding as a supervisory signal to enhance video-text alignment.

3. We introduce two synergistic strategies in TGL–**MDP** and **OSS**–which respectively decouple the learning process and refine the supervision signal to enable robust temporal understanding.

4. We establish a new state-of-the-art on the MeViS benchmark, demonstrating the effectiveness of our proposed approach.

## 2 RELATED WORKS

### 2.1 REFERRING VIDEO OBJECT SEGMENTATION

Ref-YouTube-VOS (Seo et al., 2020) and Ref-DAVIS (Yang et al., 2019) originally defined the core challenges of RVOS, where the goal is to segment objects in videos based on natural language descriptions. With the advent of transformer-based segmentation architectures (Cheng et al., 2022;

Zhu et al., 2020), many RVOS methods began to adopt query-driven frameworks to better align visual content with language. MTTR (Botach et al., 2022) leverages a Video Swin Transformer to model spatio-temporal context, while ReferFormer (Wu et al., 2022) improves grounding through multi-scale feature aggregation and text-conditioned queries. To address the limitations of early benchmarks—such as simple expressions and single-object scenarios—MeViS (Ding et al., 2023) introduces more complex descriptions involving multiple objects per video, along with a motion-aware baseline that enhances temporal modeling. Building on this, He et al. (He & Ding, 2024) proposed a method that explicitly decouples static and motion cues, further improving temporal consistency and segmentation accuracy. More recently, the rise of vision-language models (VLMs) has spurred the development of large-scale reasoning-based approaches. VISA (Yan et al., 2024), for example, utilizes Chat-UniVi (Jin et al., 2024) as a video reasoning agent to identify relevant keyframes, segments them using SAM (Kirillov et al., 2023), and propagates masks with XMem (Cheng & Schwing, 2022). Several methods (Gong et al., 2025; Lin et al., 2025) follow a similar paradigm, adopting powerful segmentation models like SAM and employing multimodal LLMs such as Chat-UniVi and LLaVA (Liu et al., 2023) as reasoning modules. SAMWISE (Cuttano et al., 2025) introduces a lightweight cross-modal adapter that enables effective text grounding within SAM2 (Ravi et al., 2024). Despite these advances, accurately identifying text-relevant keyframes remains a key bottleneck. This step is critical for robust grounding, especially in complex scenarios requiring temporal consistency and multi-object reasoning.

## 2.2 Temporal Grounding with Vision-Language Models

Pioneering works like CLIP (Radford et al., 2021) established a shared embedding space for images and text through contrastive pre-training. This was enhanced by subsequent models such as BLIP-2 (Li et al., 2023a), which efficiently bridged vision encoders with Large Language Models (LLMs). A common application of these models in video understanding is to treat a video as a collection of frames and compute frame-query relevance scores to identify key moments (Liu et al., 2025; Tan et al., 2025). To better capture temporal context, the field has evolved towards video-native Vision-Language Models (VLMs). Models like LLaMA-Vid (Li et al., 2024) and Chat-UniVi (Jin et al., 2024) introduced mechanisms such as frame compression and dynamic multi-scale tokens to process temporal sequences more effectively. Recent advancements like Qwen2.5-VL (Bai et al., 2025) further refine this by employing dynamic resolution and explicit temporal encoding. These sophisticated video-VLMs are now integral to pipelines requiring temporal reasoning, such as moment retrieval (Meinardus et al., 2024; Yan et al., 2025; Xu et al., 2025). Despite these significant advances, achieving consistent and fine-grained alignment for segmentation in complex scenarios with subtle actions remains a challenging research area.

## 3 MeViS-M Dataset

MeViS (Ding et al., 2023) poses a significant challenge for video-text alignment due to its large number of objects and the frequent use of motion-centric expressions. While the dataset offers dense annotations across diverse scenarios, it lacks explicit annotations for text-relevant moments—temporal intervals during which objects perform actions or exhibit states described by the referring sentence. This absence leads existing methods to rely on random frame sampling during training, which often includes frames irrelevant to the expression. Consequently, such indiscriminate sampling hinders effective video-text alignment, making it more difficult for models to associate the referring sentence with the correct object across time.

To address this issue, we introduce **MeViS-M** dataset, an augmented version of the MeViS dataset with explicit moment annotations that enable finer-grained video-text alignment during training. Since a referring sentence may describe specific actions or states involving one or more objects, we annotate text-relevant temporal spans on a per-object basis. For each video, we manually identify the temporal intervals during which each object is semantically relevant to the referring expression, resulting in object-specific moment sets. Formally, for each object $i$, we define a moment index set $\mathcal{M}_i \subseteq \{1, \ldots, T_V\}$, where $T_V$ is the total number of frames in the video. The union of these sets across all objects constitutes the set of text-relevant moment indices, $\mathcal{M}^+ = \bigcup_i \mathcal{M}_i$, while its complement $\mathcal{M}^- = \{1, \ldots, T_V\} \setminus \mathcal{M}^+$ representing irrelevant frames.

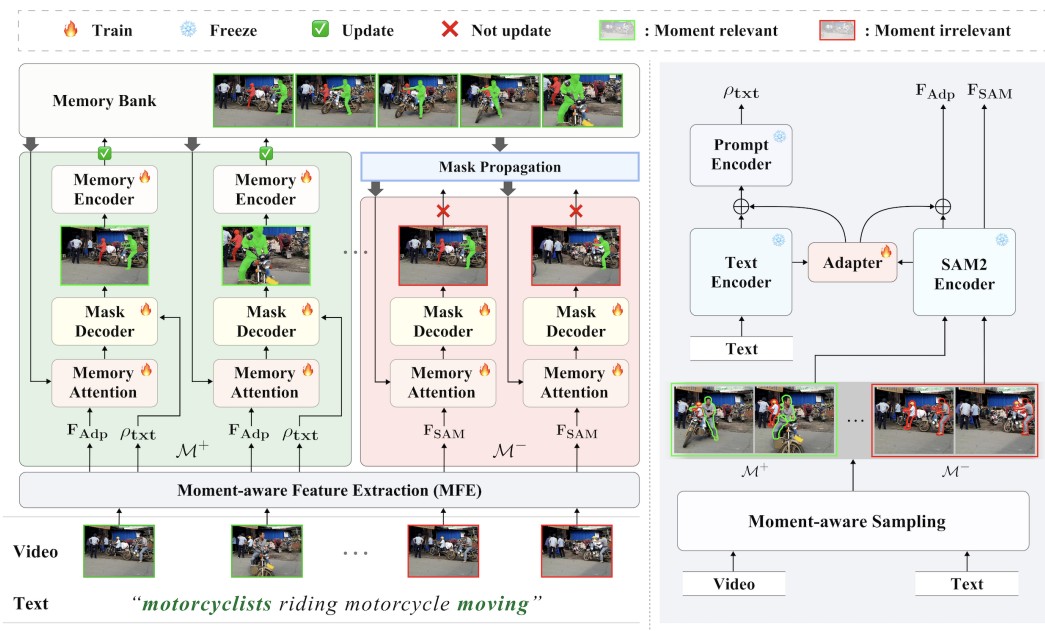

(a) Moment-guided Dual-path Propagation (MDP)     (b) Moment-aware Feature Extraction (MFE)

Figure 2: **Overall pipeline.** In (a), $\mathbf{F}_{\text{Adp}}$ of text-relevant frames are utilized for mask generation and memory update, while text-irrelevant frames employ $\mathbf{F}_{\text{SAM}}$ for mask generation without contributing to the memory update. (b) illustrates how $\mathbf{F}_{\text{Adp}}$ and $\mathbf{F}_{\text{SAM}}$ are extracted from relevant and irrelevant frames, respectively, and how visual features are integrated into the prompt.

## 4 TEMPORALLY GROUNDED LEARNING

We present *Temporally Grounded Learning (TGL)*, a novel RVOS framework designed to improve video-text alignment by explicitly leveraging text-relevant moments, as illustrated in Fig. 2. Given a video $V = \{I_t\}_{t=1}^{T_V}$ consisting of $T_V$ frames and a linguistic expression $E = \{e_l\}_{l=1}^{L}$ composed of $L$ words, the objective is to segment and temporally track the objects mentioned in the expression throughout the video. TGL achieves this through the following key components: (1) *Moment-guided Dual-path Propagation (MDP)*, which enables consistent object tracking across both text-relevant and irrelevant frames by learning propagation with a moment-centric memory bank, and (2) *Object-level Selective Supervision (OSS)*, which improves supervision by filtering out objects unrelated to the expression in the sampled clip.

### 4.1 BASELINE ARCHITECTURE

We adopt SAM2 (Ravi et al., 2024) and RoBERTa (Liu et al., 2019) for video segmentation and text encoding. To enable effective vision-language interaction, we employ a lightweight adapter module (Cuttano et al., 2025), which fuses visual and textual features through bidirectional cross-attention. Given a video $V = \{I_t\}_{t=1}^{T_V}$ and a referring expression $E = \{e_l\}_{l=1}^{L}$, we sample a short clip consisting of $T$ frames for training. We extract intermediate visual features $\mathbf{F}^k \in \mathbb{R}^{T \times H_k \times W_k \times C_k}$ from the visual encoder, and textual embeddings $\mathbf{E}^k \in \mathbb{R}^{L \times D}$ from the text encoder at the $k$-th layer. The adapter module updates both features via bidirectional cross-attention as follows:

$$\mathbf{F}_{\text{Adp}}^k = \mathbf{F}^k + h(\mathbf{F}^k \mathbf{W}_{\text{down}}^{\text{v}}, \mathbf{E}^k \mathbf{W}_{\text{down}}^{\text{t}}) \mathbf{W}_{\text{up}}^{\text{v}}, \quad \mathbf{E}_{\text{Adp}}^k = \mathbf{E}^k + h(\mathbf{E}^k \mathbf{W}_{\text{down}}^{\text{t}}, \mathbf{F}^k \mathbf{W}_{\text{down}}^{\text{v}}) \mathbf{W}_{\text{up}}^{\text{t}}, \quad (1)$$

where $h(\cdot, \cdot)$ denotes a cross-attention function. Here, $\mathbf{W}_{\text{down}}^{\text{v}}$ and $\mathbf{W}_{\text{down}}^{\text{t}}$ are learnable down-projection matrices for visual and textual features, respectively. Correspondingly, $\mathbf{W}_{\text{up}}^{\text{v}}$ and $\mathbf{W}_{\text{up}}^{\text{t}}$ are the learnable up-projection matrices. We refer to the outputs of the final (i.e., $K$-th) adapter layer as $\mathbf{F}_{\text{Adp}} := \mathbf{F}_{\text{Adp}}^K$ and $\mathbf{E}_{\text{Adp}} := \mathbf{E}_{\text{Adp}}^K$.

To construct a text prompt, we leverage the adapter-enhanced text features $\mathbf{E}_{\text{Adp}}$, where $\mathbf{E}_{\text{C}}$ denotes the contextual embedding from the `[CLS]` token, and $\mathbf{E}_{\text{M}}$ represents the motion-centric embedding extracted from verb-related tokens. These embeddings are concatenated via the function $\psi(\cdot, \cdot)$ and passed through an MLP:

$$\rho_{\text{txt}} = \text{MLP}\left(\psi(\mathbf{E}_{\text{C}}, \mathbf{E}_{\text{M}})\right). \tag{2}$$

We first compute memory-attended visual features $\mathbf{F}_{\text{mem}}$ using a memory attention module (Ravi et al., 2024), which attends over the current adaptive feature $\mathbf{F}_{\text{Adp}}$ using a memory bank $\mathcal{B}$. The resulting feature $\mathbf{F}_{\text{mem}}$ is then passed to the mask decoder $\mathcal{D}$ along with the text prompt $\rho_{\text{txt}}$ to produce soft segmentation mask $\mathbf{P} \in \mathbb{R}^{H \times W}$:

$$\mathbf{P} = \mathcal{D}(\mathbf{F}_{\text{mem}}, \rho_{\text{txt}}), \quad \mathbf{F}_{\text{mem}} = \text{MemoryAttn}(\mathbf{F}_{\text{Adp}}, \mathcal{B}). \tag{3}$$

The memory bank $\mathcal{B}$ is updated using the soft mask $\mathbf{P}$, following the strategy introduced in SAM2 (Ravi et al., 2024) to enhance temporal consistency. The prompt input $\rho_{\text{txt}}$ is optional and used only when available (e.g., during the initial grounding stage). We obtain the final binary mask via thresholding: $\hat{\mathbf{Y}} = (\mathbf{P} > 0)$.

### 4.2 Design Motivation

A natural strategy for temporally grounded RVOS is to train the model solely on text-relevant moments, denoted as $\mathcal{M}^+$. This approach directs the model's attention toward aligning visual content with the referring expression by providing supervision only on the frames that are semantically consistent with the input text. During inference, the model first performs moment retrieval to identify $\mathcal{M}^+$, segments the referred object within these frames, and subsequently propagates the segmentation masks to the remaining, non-relevant frames $\mathcal{M}^-$. Although training on the text-relevant frames $\mathcal{M}^+$ encourages the model to align visual features with the referring expression $\mathcal{E}$, this naive strategy introduces two significant challenges at inference time: (1) applying text-conditioned features to semantically irrelevant frames $\mathcal{M}^-$ can introduce misleading signals, and (2) a mismatch arises between memory features (from $\mathcal{M}^+$) and query features (from $\mathcal{M}^-$), impairing effective mask propagation.

First, conditioning all frames on $E$ is suboptimal, especially for $\mathcal{M}^-$, which lacks visual evidence corresponding to the expression. Applying the adapter to these irrelevant frames may introduce noise or ambiguity. To address this, it is more appropriate to rely on raw visual features $\mathbf{F}_{\text{SAM}}$—extracted from the frozen SAM2 encoder—without text conditioning for $\mathcal{M}^-$. Second, this asymmetric feature design leads to a feature inconsistency during inference. Specifically, the memory bank is constructed using text-conditioned features $\mathbf{F}_{\text{Adp}}$ from $\mathcal{M}^+$, while memory queries are performed using the unconditioned features $\mathbf{F}_{\text{SAM}}$ from $\mathcal{M}^-$. However, since the model is trained exclusively on $\mathcal{M}^+$, it never learns to conduct memory attention using unconditioned query features like $\mathbf{F}_{\text{SAM}}$. This discrepancy between memory and query representations degrades the model's ability to propagate masks accurately, ultimately resulting in poor tracking performance.

### 4.3 Moment-guided Dual-path Propagation

We propose *Moment-guided Dual-path Propagation (MDP)*, a strategy designed to improve both semantic grounding and consistent tracking by selectively leveraging frames within and outside the referred moment. At the core of MDP is a *moment-aware feature extraction (MFE)* strategy that first partitions each video into text-relevant ($\mathcal{M}^+$) and text-irrelevant ($\mathcal{M}^-$) segments using either moment annotations or a retrieval model. By handling $\mathcal{M}^+$ and $\mathcal{M}^-$ differently, MDP encourages the model to jointly learn cross-modal alignment and temporal consistency.

For frames in the text-relevant segment $\mathcal{M}^+$, we apply a cross-modal adapter to extract text-conditioned features $\mathbf{F}_{\text{Adp}}$. In contrast, for frames in the text-irrelevant segment $\mathcal{M}^-$, we use the raw visual features $\mathbf{F}_{\text{SAM}}$ obtained from the frozen SAM2 encoder to prevent semantic contamination from unrelated textual input. To support memory-based reasoning, we compute memory-attended features for both segments using a shared memory attention module and a memory bank $\mathcal{B}$ as follows:

$$\mathbf{F}_{\text{mem}}^+ = \text{MemoryAttn}(\mathbf{F}_{\text{Adp}}, \mathcal{B}), \quad \mathbf{F}_{\text{mem}}^- = \text{MemoryAttn}(\mathbf{F}_{\text{SAM}}, \mathcal{B}). \tag{4}$$

The decoder $\mathcal{D}$ then predicts the soft segmentation mask $\mathbf{P}$ by conditioning on the appropriate memory-attended feature. For $\mathcal{M}^+$, the decoder also receives the text prompt $\rho_{\text{txt}}$; for $\mathcal{M}^-$, only visual information is used:

$$\mathbf{P} = \begin{cases} \mathcal{D}(\mathbf{F}_{\text{mem}}^+, \rho_{\text{txt}}), & \text{if } t \in \mathcal{M}^+, \\ \mathcal{D}(\mathbf{F}_{\text{mem}}^-), & \text{if } t \in \mathcal{M}^-. \end{cases} \tag{5}$$

This dual-pathway design enables the model to ground the expression precisely within $\mathcal{M}^+$, while relying on memory-guided propagation for consistent segmentation across $\mathcal{M}^-$.

To enable robust tracking beyond the text-relevant segment $\mathcal{M}^+$, we adopt the memory bank management strategy from SAM2 (Ravi et al., 2024), with a key modification: the memory bank stores features exclusively from $\mathcal{M}^+$. This design choice ensures that all memory entries used for guidance are semantically grounded by the referring expression. During inference, when processing frames in the text-irrelevant segment $\mathcal{M}^-$, the model retrieves memory features from $\mathcal{M}^+$ to assist segmentation. By attending to these reliable, text-aligned memory representations—rather than relying on local frame-to-frame continuity—the model is better equipped to track objects across ambiguous or visually uninformative regions. This memory-driven mechanism reduces error accumulation and significantly improves the temporal stability and long-range consistency of the segmentation masks.

### 4.4 Object-level Selective Supervision

Conventional RVOS methods supervise the model using ground-truth (GT) masks for all objects visible in the sampled frames, regardless of whether they are semantically relevant to the referring expression. However, in moment-aware scenarios, each object $i$ is associated with a distinct text-relevant moment $\mathcal{M}_i$, which specifies the temporal span during which the object is referred to by the expression $E$. Consequently, sampled frames may include objects that are visually present but not semantically aligned with the text query, introducing supervisory noise and weakening cross-modal alignment.

To mitigate this issue, we introduce an object-filtered supervision strategy called *Object-level Selective Supervision (OSS)*, which leverages object-wise moment annotations to selectively supervise only the relevant objects in each training clip. Given a sampled frame index set $\mathcal{T}$ of length $T$, we discard the GT masks of any object $i$ whose annotated moment $\mathcal{M}_i$ does not overlap with $\mathcal{T}$, i.e., when $\mathcal{T} \cap \mathcal{M}_i = \emptyset$. We retain only those masks $\mathbf{Y}_t^i \in \mathbb{R}^{H \times W}$ for which the condition $\mathcal{T} \cap \mathcal{M}_i \neq \emptyset$ holds. The resulting filtered set of ground-truth masks is defined as:

$$\mathbf{Y}_{\text{OSS}} = \bigcup_{t \in \mathcal{T}} \left\{ \mathbf{Y}_t^i \mid \mathcal{T} \cap \mathcal{M}_i \neq \emptyset \right\}. \tag{6}$$

We use this filtered mask set $\mathbf{Y}_{\text{OSS}}$ to supervise the model's predictions $\mathbf{P}_{\mathcal{T}}$ over the sampled frame interval $\mathcal{T}$. The final training loss combines the Dice loss $\mathcal{L}_{\text{Dice}}$ and Focal loss $\mathcal{L}_{\text{Foc}}$ (Lin et al., 2017) as follows:

$$\mathcal{L} = \lambda_{\text{Dice}} \mathcal{L}_{\text{Dice}}(\mathbf{Y}_{\text{OSS}}, \mathbf{P}_{\mathcal{T}}) + \lambda_{\text{Foc}} \mathcal{L}_{\text{Foc}}(\mathbf{Y}_{\text{OSS}}, \mathbf{P}_{\mathcal{T}}). \tag{7}$$

By aligning supervision targets with the temporal scope of the referring expression, *Object-level Selective Supervision* effectively filters out semantically irrelevant objects, thereby improving the precision of cross-modal learning and enhancing segmentation performance in RVOS.

## 5 Experiments

We conduct extensive experiments to validate our core hypothesis: leveraging temporal grounding as a direct learning signal is crucial for complex RVOS tasks. Our evaluation is primarily performed on the challenging MeViS dataset (Ding et al., 2023), with additional zero-shot evaluations on Ref-YouTube-VOS (Seo et al., 2020) and Ref-DAVIS (Khoreva et al., 2019) to assess generalization. All models are evaluated using the standard mean Intersection-over-Union and Contour Accuracy ($\mathcal{J}\&\mathcal{F}$) metric. Our analysis is structured as follows: we first present the main results on MeViS, then conduct detailed ablation studies to verify the effectiveness of our components, followed by an in-depth analysis of various temporal grounding strategies for inference, and finally, we report the model's generalization capabilities.

Table 1: Comparison on the MeViS dataset. **Oracle** uses ground-truth moments from MeViS-M at inference. † indicates methods that leverage vision-language models (VLMs) for keyframe selection. **Oracle** + Ours† uses VLMs to extract keyframes from ground-truth moments. We adopt Chrono (Meinardus et al., 2024) and BLIP-2 (Li et al., 2023a) as keyframe selectors.

| Method | Visual Encoder | Text Encoder | Total Params | $\mathcal{J}\&\mathcal{F}$ | $\mathcal{J}$ | $\mathcal{F}$ |
|---|---|---|---|---|---|---|
| ***Large VLM* based** | | | | | | |
| LISA (Lai et al., 2024) [CVPR'24] | ViT-H | LLaVa | 7 B | 37.2 | 35.1 | 39.4 |
| VISA† (Yan et al., 2024) [ECCV'24] | ViT-H | Chat-UniVi | 7 B | 43.5 | 40.7 | 46.3 |
| VideoLISA (Bai et al., 2024) [NIPS'24] | ViT-H | Phi-3 | 3.8 B | 42.3 | 39.4 | 45.2 |
| DTOS† (Tian et al., 2025) [CVPR'25] | Hiera-L | LLaMA | 9 B | 48.9 | 45.2 | 52.6 |
| VRS-HQ† (Gong et al., 2025) [CVPR'25] | Hiera-L | Chat-UniVi | 7 B | 50.6 | 47.6 | 53.7 |
| GLUS† (Lin et al., 2025) [CVPR'25] | Hiera-L | LLaVa | 7 B | 51.3 | 48.5 | 54.2 |
| ***Regular Methods*** | | | | | | |
| MTTR (Botach et al., 2022) [CVPR'22] | V-Swin-T | RoBERTa | - | 30.0 | 28.8 | 31.2 |
| ReferFormer (Wu et al., 2022) [CVPR'22] | V-Swin-B | RoBERTa | 237 M | 31.0 | 29.8 | 32.2 |
| OnlineRefer (Wu et al., 2023) [ICCV'23] | Swin-L | RoBERTa | 232 M | 32.3 | 31.5 | 33.1 |
| LMPM (Ding et al., 2023) [ICCV'23] | Swin-T | RoBERTa | 195 M | 37.2 | 34.2 | 40.2 |
| DsHmp (He & Ding, 2024) [CVPR'24] | Swin-T | RoBERTa | 272 M | 46.4 | 43.0 | 49.8 |
| SAMWISE (Cuttano et al., 2025) [CVPR'25] | Hiera-B | RoBERTa | 202 M | 49.5 | 46.6 | 52.4 |
| SAMWISE† (Cuttano et al., 2025) [CVPR'25] | Hiera-B | RoBERTa | 202 M | 48.9 | 46.0 | 51.7 |
| Ours† | Hiera-B | RoBERTa | 202 M | **51.6** | **48.8** | **54.5** |
| ***Oracle*** | | | | | | |
| SAMWISE (Cuttano et al., 2025) [CVPR'25] | Hiera-B | RoBERTa | 202 M | 50.3 | 47.4 | 53.3 |
| GLUS (Lin et al., 2025) [CVPR'25] | Hiera-L | LLaVa | 7B | 51.5 | 48.7 | 54.3 |
| Ours | Hiera-B | RoBERTa | 202 M | 53.5 | 50.7 | 56.3 |
| Ours | Hiera-L | RoBERTa | 355 M | 55.1 | 52.6 | 57.7 |
| Ours† | Hiera-B | RoBERTa | 202 M | 54.8 | 52.1 | 57.5 |
| Ours† | Hiera-L | RoBERTa | 355 M | **55.7** | **52.9** | **58.5** |

Our model is built upon SAM2 (Ravi et al., 2024) as the base segmentation framework, with Hiera-Base and Hiera-Large (Ryali et al., 2023) serving as the visual backbones. We use RoBERTa (Liu et al., 2019) as the text encoder. Further implementation details are described in Sec. A.3.

## 5.1 Main Results on MeViS

As shown in Table 1, we compare TGL against state-of-the-art methods in two settings: a **Regular** setting using a VLM for moment prediction, and an **Oracle** setting. This **Oracle** setting is designed to isolate the model's upper-bound capability by providing it with the ground-truth moment annotations from MeViS-M directly at inference time.

In the standard **Regular** setting, our TGLframework with a Hiera-B backbone already establishes a new state-of-the-art with **51.6** $\mathcal{J}\&\mathcal{F}$, outperforming the strongest comparable method, SAMWISE, by a margin of **+2.1** points. The importance of our training paradigm becomes even clearer in the **Oracle** setting. When compared against SAMWISE, which uses an identical backbone, our model's score surges to **53.5** $\mathcal{J}\&\mathcal{F}$, widening the performance gap over SAMWISE (50.3) to **+3.2** points. The limitation of conventional methods is most apparent with GLUS, a massive 7B parameter model. Given GT moments, its score improves only marginally from 51.3 to 51.5 (+0.2), proving that large-scale pre-training is insufficient for utilizing temporal information without explicit training. In contrast, our TGL with a Hiera-L backbone achieves **55.1** in the same setting, surpassing GLUS by a significant **+3.6** points and highlighting the profound impact of our learning strategy.

## 5.2 Ablation Study

To purely evaluate their effectiveness in leveraging temporal signals for video-text alignment, all experiments in this section are performed under the **Oracle** setting. This controlled setup removes the influence of an external moment retrieval model, allowing for a direct assessment of how our framework utilizes the learning signal. All experiments use a Hiera-B backbone and report the $\mathcal{J}\&\mathcal{F}$ score on the MeViS 'valid_u' set.

Table 2: Ablation study on main components.

| Sampling from MeViS-M | MDP | OSS | $\mathcal{J}\&\mathcal{F}$ |
|:---:|:---:|:---:|:---:|
| | | | 56.8 |
| ✓ | | | 58.0 |
| ✓ | ✓ | | 59.4 |
| ✓ | | ✓ | 58.3 |
| ✓ | ✓ | ✓ | **60.8** |

Table 3: Ablation study on MDP.

| MFE | Memory bank | $\mathcal{J}\&\mathcal{F}$ |
|:---:|:---:|:---:|
| | All frames | 58.0 |
| | $\mathcal{M}^+$ | 58.9 |
| ✓ | All frames | 60.3 |
| ✓ | $\mathcal{M}^+$ | **60.8** |

**Ablation study on main components.** Our method integrates three core components: moment-aware sampling based on MeViS-M, Moment-guided Dual-path Propagation (MDP), and Object-level Selective Supervision (OSS). As shown in Tab. 2, training without any moment-aware design yields a score of 56.8. Incorporating moment-aware sampling via MeViS-M provides a modest improvement to 58.0. Adding either MDP or OSS individually yields further gains, reaching 59.4 and 58.3, respectively. When all components are combined, our model achieves 60.8, confirming that each contributes complementarily to the final performance. Notably, this represents a significant improvement of **+4.0** over the baseline, highlighting the effectiveness of our temporal formulation.

**Ablation study on MDP.** Tab. 3 analyzes two core design elements within MDP: (1) Moment-aware Feature Enhancement (MFE), and (2) selective memory construction using only $\mathcal{M}^+$ frames. Without MFE and using a memory bank built from all frames, the model achieves 58.0. Restricting the memory bank to $\mathcal{M}^+$ improves the performance to 58.9. Applying MFE alone yields a larger gain of 60.3, and combining both MFE and selective memory results in the highest score of 60.8. These results validate that both components are critical to achieving accurate propagation.

## 5.3 Analysis on Temporal Grounding

We conduct a detailed analysis to evaluate the efficacy of various temporal grounding strategies for inference. This analysis underscores the limitations of existing models in temporal localization and demonstrates how providing a better temporal signal consistently improves performance.

**Analysis of VLM-based Grounding.** As shown in Tab. 4-(a), a significant performance gap exists between 'valid_u' and 'valid' sets, which can be attributed to the VLMs' temporal localization capabilities. Most VLMs achieve a Top1 Acc above 65% on 'valid_u', but this drops to around 50% on the more challenging 'valid' set. This indicates a weakness in identifying the correct moment in complex scenarios, as shown in Fig. 6. Furthermore, the table reveals that the highest Top1 Accuracy does not guarantee the best RVOS performance. For instance, LLaMA-VID (Li et al., 2024), a common choice in prior work, achieves the highest $\mathcal{J}\&\mathcal{F}$ score on 'valid_u' (**59.3**) but has a low Top1 Acc. This suggests that simply selecting a VLM based on its grounding accuracy is a suboptimal strategy. We chose BLIP-2 (Li et al., 2023a) for further experiments due to its strong and balanced performance across both metrics and splits. Tab. 4-(b) shows that selecting the top-4 keyframes via BLIP-2 improves RVOS performance, confirming that using a small set of relevant frames is more effective than relying on a single, potentially noisy one.

**Analysis of Hybrid Grounding.** Identifying an initial high-quality temporal span is crucial for enhancing RVOS performance. To this end, we adopt a Hybrid approach that first predicts the initial temporal span and then selects the most relevant frames within it. In (Tab. 4-(c)), we use Chrono(Meinardus et al., 2024), a moment retrieval model trained on MeViS-M, to identify the initial span, and subsequently apply BLIP-2 to select the top-$k$ frames. This strategy boosts the 'valid' score to **51.6** $\mathcal{J}\&\mathcal{F}$ (at $k = 4$), a substantial improvement(**+2.7**) over the VLM-only method. This trend is further amplified in the Oracle Hybrid setting. As shown in Tab. 4-(d), simply providing the oracle moment span (Baseline) yields 53.5 $\mathcal{J}\&\mathcal{F}$ on 'valid'. By further using a VLM to pinpoint the most relevant frame within this perfect interval, performance is consistently improved, with Chat-UniVi (Jin et al., 2024) reaching **54.6**. Given BLIP-2's balanced performance, we conducted an additional top-$k$ analysis (Tab. 4-(e)). The results show a peak performance of **54.8** at $k = 8$, with $k = 4$ also providing a strong balance. These experiments collectively prove that the quality of the temporal signal is the most critical factor for performance, validating our core hypothesis.

Table 4: Analysis of temporal grounding strategies. (a) Temporal grounding top1 accuracy and $\mathcal{J}\&\mathcal{F}$ scores of various VLMs. (b) Impact of varying the number of top-$k$ keyframes selected by BLIP-2. (c) Hybrid approach combining Chrono with BLIP-2. (d) Hybrid approaches combining Oracle moment with various VLMs. (e) Hybrid approach combining Oracle moment with BLIP-2.

(a) Analysis on temporal grounding with various VLMs.

| Method | valid_u | | valid | |
|---|---|---|---|---|
| | Top1 Acc | $\mathcal{J}\&\mathcal{F}$ | Top1 Acc | $\mathcal{J}\&\mathcal{F}$ |
| BLIP-2 | **73.5** | 58.6 | 49.9 | 48.4 |
| CLIP | 65.0 | 58.3 | 52.6 | **48.8** |
| Unmasked Teacher | 64.7 | 58.6 | **55.0** | 48.4 |
| LLaMA-VID | 65.5 | **59.3** | 42.6 | 47.7 |
| Chat-UniVi | 61.8 | 57.5 | 47.5 | 48.4 |

(b) Analysis on BLIP-2

| top-$k$ | valid_u | valid |
|---|---|---|
| 1 | 58.6 | 48.4 |
| 2 | 59.8 | 48.7 |
| 4 | **60.7** | **48.9** |
| 8 | 60.0 | 48.8 |
| 16 | 60.3 | **48.9** |
| 32 | 60.6 | 48.8 |

(c) Chrono + Blip-2.

| top-$k$ | valid_u | valid |
|---|---|---|
| 1 | 58.6 | **51.6** |
| 2 | 59.2 | 51.5 |
| 4 | 60.1 | **51.6** |
| 8 | 59.7 | 51.2 |
| 16 | 59.8 | 51.2 |
| 32 | **60.2** | 51.3 |

(d) Oracle + various VLMs.

| Method | valid_u | valid |
|---|---|---|
| Baseline (Oracle) | 60.8 | 53.5 |
| + BLIP-2 | **62.0** | 54.4 |
| + CLIP | 61.7 | 54.1 |
| + Unmasked Teacher | 61.5 | 53.7 |
| + LLaMA-VID | **62.0** | 53.9 |
| + Chat-UniVi | 61.3 | **54.6** |

(e) Oracle + BLIP-2.

| top-$k$ | valid_u | valid |
|---|---|---|
| 1 | 62.0 | 54.4 |
| 2 | 62.3 | 54.6 |
| 4 | **62.9** | 54.5 |
| 8 | 62.3 | **54.8** |
| 16 | 62.3 | 54.6 |
| 32 | 62.7 | 54.6 |

## 5.4 GENERALIZATION TO OTHER DATASETS

We also evaluate the generalization ability of our model trained on the MeViS dataset by testing its zero-shot performance on the validation set of two other RVOS benchmarks: Ref-YouTube-VOS (Seo et al., 2020) and Ref-DAVIS (Khoreva et al., 2019). As shown in Tab. 5, our method achieves strong zero-shot results, out-performing recent state-of-the-art approaches such as DsHmp (He & Ding, 2024) and SAMWISE (Cuttano et al., 2025) by a notable margin. In particular, our model surpasses DsHmp by +14.5 on Ref-YouTube-VOS and +2.7 on Ref-DAVIS, and outperforms SAMWISE by +4.2 and +2.0 on the respective benchmarks. This highlights the effectiveness of moment-aware training in capturing generalized language-visual alignment across diverse domains.

Table 5: Comparison of zero-shot performance.

| Model | Backbone | Ref-YouTube-VOS | Ref-DAVIS |
|---|---|---|---|
| DsHmp | Swin-T | 45.8 | 64.7 |
| SAMWISE | Hiera-B | 56.1 | 65.4 |
| Ours | Hiera-B | **60.3** | **67.4** |

## 6 CONCLUSION

In this work, we addressed a fundamental limitation in Referring Video Object Segmentation: the absence of an explicit temporal learning signal, which leads to flawed, semantically contradictory supervision in conventional training paradigms. To rectify this, we introduced **MeViS-M**, a new dataset to provide object-level temporal annotations on the challenging MeViS benchmark, thereby supplying this critical missing signal. We then proposed the **Temporally Grounded Learning (TGL)** framework, a novel learning paradigm designed to effectively leverage this signal. TGL incorporates two synergistic strategies: **Moment-guided Dual-path Propagation (MDP)**, which decouples the learning process based on temporal relevance, and **Object-level Selective Supervision (OSS)**, which refines the supervision signal to eliminate semantic noise. Our extensive experiments demonstrate that by treating temporal grounding as a primary learning signal, our framework establishes a new state-of-the-art on MeViS. This result underscores the critical importance of explicit temporal supervision for genuine video-text alignment and paves the way for future research into more robust and temporally-aware video understanding models.

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

# A APPENDIX

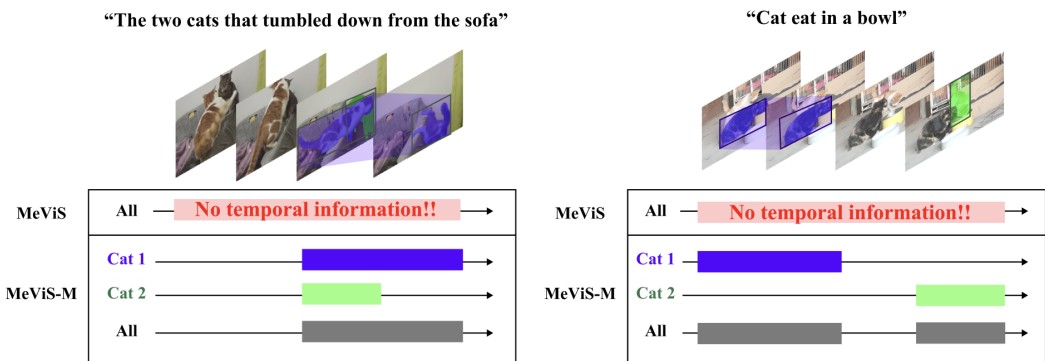

Figure 3: Moment annotation examples of MeViS-M.

## A.1 MOMENT LABEL COLLECTION

To construct the MeViS-M dataset, we manually annotate text-relevant temporal segments on top of the existing MeViS dataset (Ding et al., 2023), which consists of three splits: *train* (1,662 videos), *valid_u* (50 videos), and *valid* (140 videos). Since the *valid* split lacks mask ground-truths, we annotate expression-relevant frames only at the video level, rather than assigning object-specific moment spans. In contrast, for *train* and *valid_u*, we provide detailed moment annotations for each referred object, as shown in Fig. 3.

The annotation process is carried out by approximately 20 annotators, who manually inspect and label each video. During this process, we remove any training samples where the referred objects do not have valid mask annotations, resulting in the removal of 66 videos and 1,278 expressions from the training set. We also make several corrections to the original labels, including adding missing objects that are described in the text but absent from the annotations, and fixing cases of label ID switching. Examples of such corrections are illustrated in Fig. 4, categorized by case type. Figure 4-(a) involves partial segmentation where the target object is incompletely annotated, and moreover, the MeViS dataset lacks suitable masks to correct the ground truth; therefore, this case was excluded from the dataset. Figure 4-(b) involves a frame in which the mask IDs of two turtles are mistakenly swapped, necessitating correction of the ID assignment. Figure 4-(c) refers to instances where multiple elephants matching the referring expression are present, but only a subset are labeled in the ground-truth; thus, the missing objects are added to the annotations. Conversely, Fig. 4-(d) describes situations where objects not corresponding to the expression are annotated, and these irrelevant masks are subsequently removed. Finally, Fig. 4-(e) combines the issues from Fig. 4-(c) and Fig. 4-(d), resulting in comprehensive corrections across the affected frames.

## A.2 FURTHER ANALYSIS OF TEMPORAL GROUNDING

In this section, we provide further analysis on temporal grounding. We explore three main aspects: (1) the performance of temporal grounding using a moment retrieval model, (2) the design of a method for temporal grounding without relying on external models, and (3) a comparison of different keyframe selection methods.

### A.2.1 TEMPORAL GROUNDING WITH MOMENT RETRIEVAL MODEL

Due to the limited video-text alignment capabilities of standard VLMs, we employ Chrono (Meinardus et al., 2024), a state-of-the-art moment retrieval

Table 6: Performance metrics on different validation splits.

| Split | R1@.5 | R1@.7 | mAP | mAP@.5 | mAP@.75 | $\mathcal{J}\&\mathcal{F}$ |
|---|---|---|---|---|---|---|
| *valid_u* | 74.7 | 61.8 | 62.1 | 72.8 | 60.0 | 57.6 |
| *valid* | 61.3 | 53.0 | 52.2 | 60.6 | 51.2 | 50.4 |

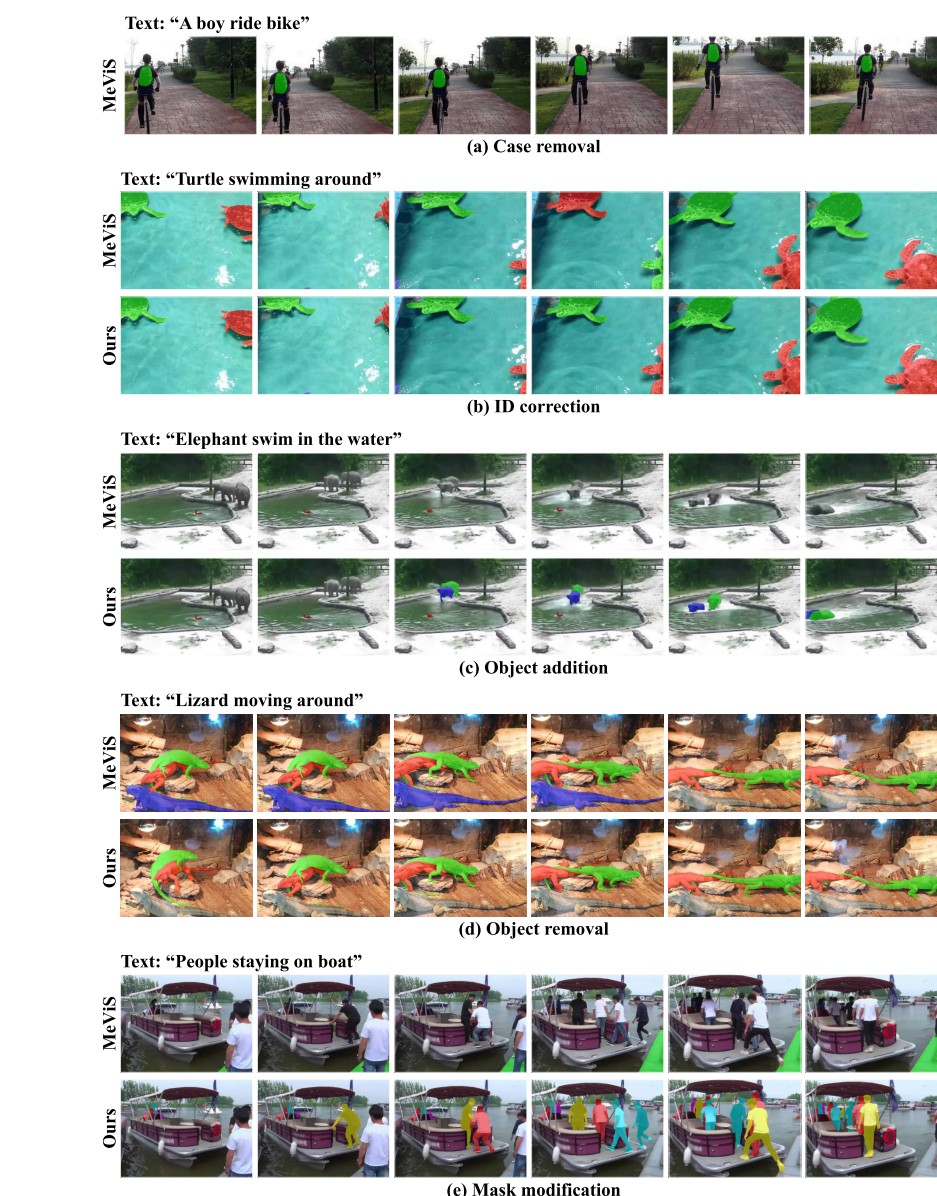

Figure 4: Examples of GT refinement in MeViS-M compared with MeViS.

model, to predict relevant temporal segments. As shown in Tab. 6, Chrono achieves an R1@.7 score of 61.8 and mAP of 62.1 on the *valid_u* and R1@.7 score of 53.0 and mAP of 52.2 on the *valid*, respectively. Using Chrono's predicted moments for inference, our model attains a $\mathcal{J}\&\mathcal{F}$ score of 57.6 on *valid_u* and 50.4 on *valid*. These results highlight the importance of accurate moment localization for achieving effective moment-aware video-text alignment in RVOS.

### A.2.2 TEMPORAL GROUNDING WITHOUT EXTERNAL VLMS

While we have shown that our TGL framework can significantly improve video-text alignment by leveraging a temporal learning signal, a potential limitation is its reliance on external models like VLMs for inference. We therefore explore a self-contained approach by designing a Temporal Alignment Module (TAM) to predict text-relevant intervals without external dependencies.

Table 7: Comparison with SAMWISE.

| Method | val_u | val |
|---|---|---|
| SAMWISE (Cuttano et al., 2025) | 55.5 | 49.5 |
| **Ours (TAM)** | **59.9** | **50.0** |

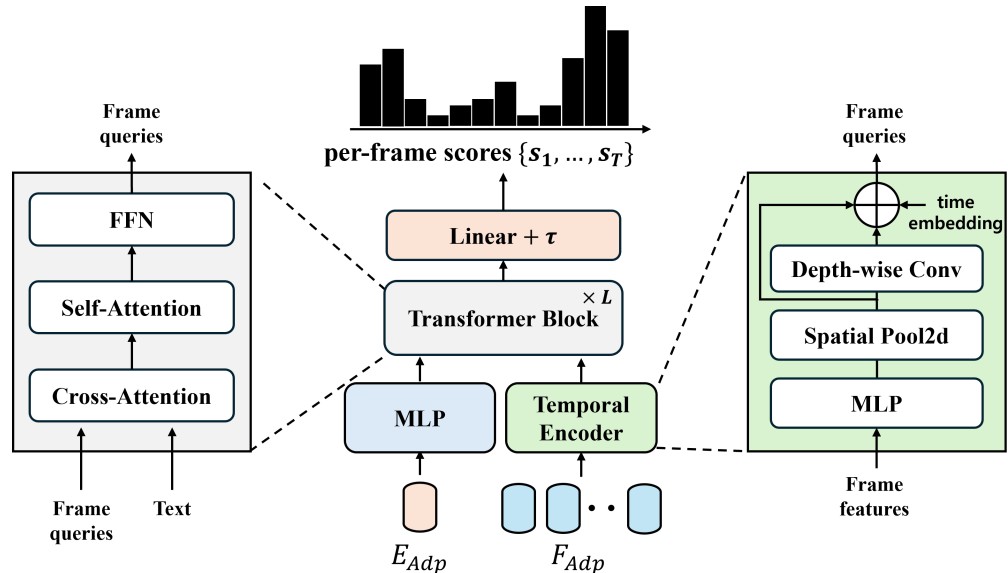

Figure 5: Illustration of Temporal Alignment Module (TAM).

As shown in Fig. 5, the TAM takes frame features $\mathbf{F}_{\text{Adp}}$ and text features $\mathbf{E}_{\text{Adp}}$ as input and outputs a relevance score for each frame. First, the frame features are fed into a Temporal Encoder to obtain frame queries. Motivated by AdaTAD (Liu et al., 2024), we adopt depth-wise convolutions to capture temporal patterns from the frame features. These frame queries are then processed through a Transformer block along with the text features. Finally, the per-frame relevance scores are computed via a linear layer and a learnable temperature parameter $\tau$: $\{s_t\}_{t=1}^{T} = \frac{1}{\tau} \times \text{Linear}(\cdot)$.

To verify the effectiveness of TAM, we further compare our model with the baseline SAMWISE. As shown in Table 7, our approach achieves consistent improvements: **+4.4** on 'valid_u' and **+0.5** on the 'valid', respectively. Furthermore, as shown in Fig. 6, the top-1 prediction results of our TAM are comparable to those of other VLMs. This demonstrates the potential of our lightweight TAM to achieve competitive performance without relying on external VLMs.

### A.2.3 COMPARISON OF KEY FRAME SELECTION METHODS

Figure 6 illustrates the Top1 predictions of various key frame selection methods alongside the MeViS-M annotations. In the first and second examples, some VLM models predict a moment that closely matches the annotated interval, demonstrating promising performance for relatively straightforward queries. However, in the third example, the predictions from VLM models are widely scattered across the timeline, failing to localize the annotated moment accurately. This highlights the limitations of current methods in achieving precise temporal localization, especially for complex or ambiguous queries, and demonstrates the need for improved text-video alignment in future models.

### A.3 IMPLEMENTATION DETAILS

**Dataset.** We train our model using moment guidance from the MeViS-M dataset, which consists of 27,292 motion-focused expressions across 1,596, 50, and 140 videos in the *train*, *valid_u*, and *valid* split, respectively. For evaluation, we test on both the *valid_u* and *valid* splits of MeViS-M. Additionally, we assess the generalization capability of our model in a zero-shot setting on two external benchmarks, including Ref-YouTube-VOS (Seo et al., 2020) and Ref-DAVIS17 (Khoreva et al., 2019). Ref-YouTube-VOS augments the original YouTube-VOS dataset (Seo et al., 2020) with approximately 15K referring expressions over 3,978 videos, and is split into 3,471 videos for training, 202 for validation, and 305 for testing. Ref-DAVIS17 extends the DAVIS17 (Pont-Tuset et al., 2017) dataset with 1.5K referring expressions annotated across 90 videos (60 for training and 30 for testing), and features high-resolution masks and complex multi-object scenarios.

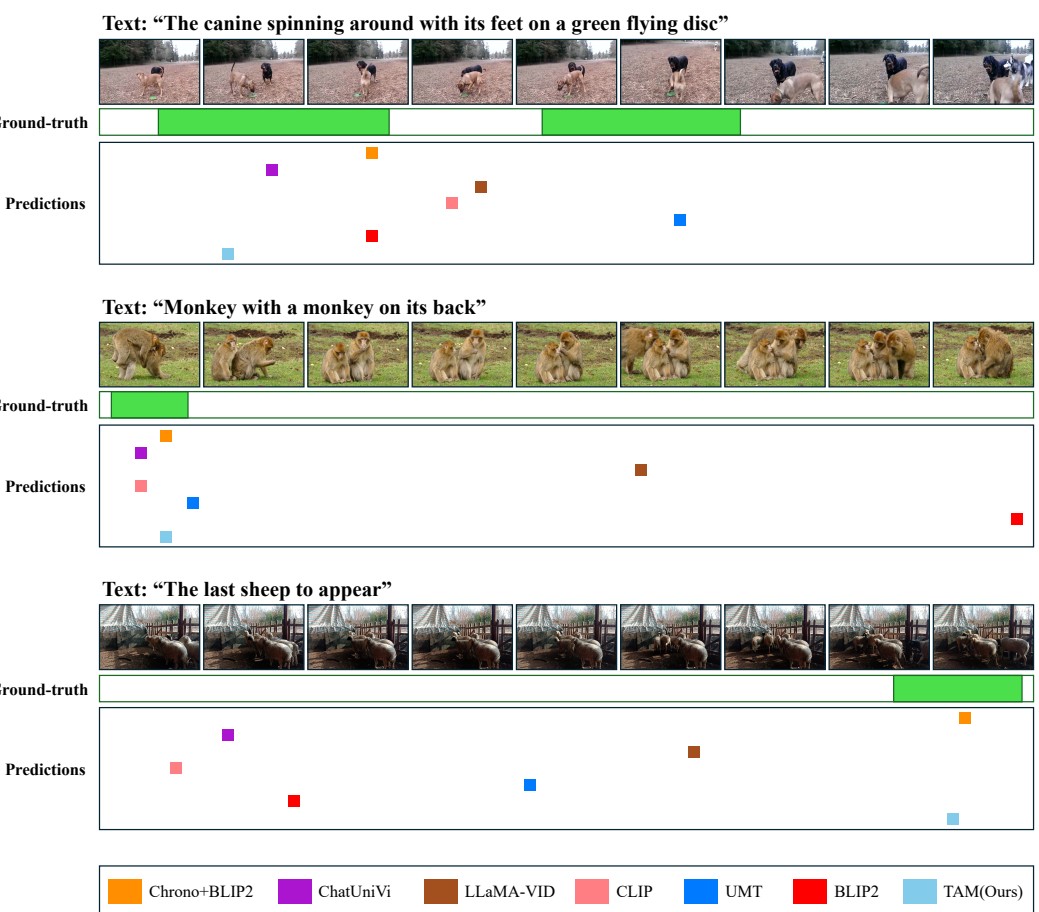

Figure 6: Moment ground-truth in MeViS-M and Top-1 predictions from VLMs and TAM

**RVOS Model.** Our model is built upon SAM2 (Ravi et al., 2024) as the video segmentation framework, with Hiera-Base and Hiera-Large (Ryali et al., 2023) used as visual backbones. For text encoding, we use RoBERTa (Liu et al., 2019), keeping both the image and text encoders frozen during training. Only the cross-modal adapter, mask decoder, and memory modules are updated, resulting in approximately 17M trainable parameters–accounting for just 8.4% of the total parameters. We utilize features from the last three stages of both encoders for the cross-modal adapter module. We sample 8 frames per video for Hiera-Base and 6 frames for Hiera-Large. To ensure temporal consistency, half of the frames are always sampled from the text-relevant segment ($\mathcal{M}^+$), while the other half are drawn from either $\mathcal{M}^+$ or irrelevant segments ($\mathcal{M}^-$). Following prior works (Wu et al., 2022; Han et al., 2023; Wu et al., 2023; Luo et al., 2024; Cuttano et al., 2025), we pretrain the model on RefCOCO/+/g (Nagaraja et al., 2016; Yu et al., 2016) for 6 epochs. Final training is conducted on MeViS-M for 1 epoch with a batch size of 4, using the Adam optimizer and a learning rate of $1\times10^{-5}$. Only features from the $\mathcal{M}^+$ segment are stored in the memory bank. When performing memory attention, the model attends to features from the 6 nearest frames within the memory. All experiments are conducted on 4 NVIDIA A100 GPUs with 40GB of memory.

**VLMs.** For inference, previous works (Yan et al., 2024; Lin et al., 2025) exploit keyframe selection using VLMs (Li et al., 2024; Jin et al., 2024). For our experiments, we evaluate five VLMs: BLIP-2 (Li et al., 2023a), CLIP (Radford et al., 2021), Unmasked Teacher (UMT) (Li et al., 2023b), LLaMa-VID (Li et al., 2024), and Chat-UniVi (Jin et al., 2024). For BLIP-2, we use a ViT-G (Fang et al., 2023) backbone. For CLIP, we use a ViT-L/14x336 backbone. For UMT, we use a ViT-L/14 backbone pretrained on 25 million image/video–text pairs. For these three models, we compute frame-wise text similarity and select the top-$k$ frames. For LLaMa-VID and Chat-UniVi, we use the same keyframe selection method as in (Yan et al., 2024; Lin et al., 2025).

Figure 7: PCA-based **feature maps** and **segmentation results** of SAMWISE & TGL.

**Moment Retrieval Model.** We employ Chrono (Meinardus et al., 2024) model for moment retrieval, which uses a BLIP-2 backbone with ViT-G (Fang et al., 2023) as the vision encoder and T5-XL (Chung et al., 2024) as the text encoder. For fine-tuning, we upsample MeViS-M samples by a factor of three when their ground-truth moments cover less than 90% of frames, excluding those spanning 90–100%. Detailed hyperparameters follow the Charades-STA (Gao et al., 2017) dataset configuration, with uniform sampling of 40 frames per video during training and 60 frames during inference. We initialize the model with pretrained weights of QVHighlights dataset (Lei et al., 2021) and finetune for five epochs on four NVIDIA A100 GPUs, with a batch size of one per GPU and gradient accumulation over eight iterations.

## A.4 ADDITIONAL RESULTS

### A.4.1 FEATURE VISUALIZATIONS

Figure 7 presents a comparative analysis of feature maps produced by the SAMWISE and TGL, with visualizations obtained using PCA for dimensionality reduction. The feature representation produced by SAMWISE lacks clear localization and fails to form distinct activations corresponding to the target object described in the expression. This indicates that the model struggles to align regions with the referring text, due to the absence of moment-aware understanding and fine-grained object-level supervision. In contrast, TGL exhibits more focused and semantically meaningful features that accurately highlight the referred object (i.e., dark regions in the **feature maps** of Fig. 7), demonstrating successful visual-text alignment. We attribute this improvement to our moment-aware strategy (MDP) and the use of object-level selective supervision (OSS), which jointly guide the model to identify the most relevant frames and learn discriminative, text-aligned representations at the object level. These results clearly show that effective temporal grounding and object-aware training significantly enhance the model's ability to align visual features with linguistic expressions.

### A.4.2 QUALITATIVE RESULTS

Figure 8 presents qualitative comparisons between our proposed model, TGL, and existing state-of-the-art RVOS models, SAMWISE and GLUS, on the MeViS dataset. In the first example, SAMWISE incorrectly segments both monkeys in response to the expression "jumping to left," indicating a lack of moment-aware reasoning. GLUS initially segments the correct object, but fails to maintain consistency, eventually highlighting an unrelated regions. In contrast, TGL accurately segments the referred object throughout the entire sequence by leveraging its moment-aware design and object-level supervision. In the remaining two examples, both SAMWISE and GLUS continue to rely on static appearance cues—such as the presence of a monkey or a rabbit—rather than understanding the motion described in the expression. These results demonstrate that, unlike prior models, TGL effectively grounds language to visual motion by incorporating a moment-aware approach and fine-grained object-level guidance.

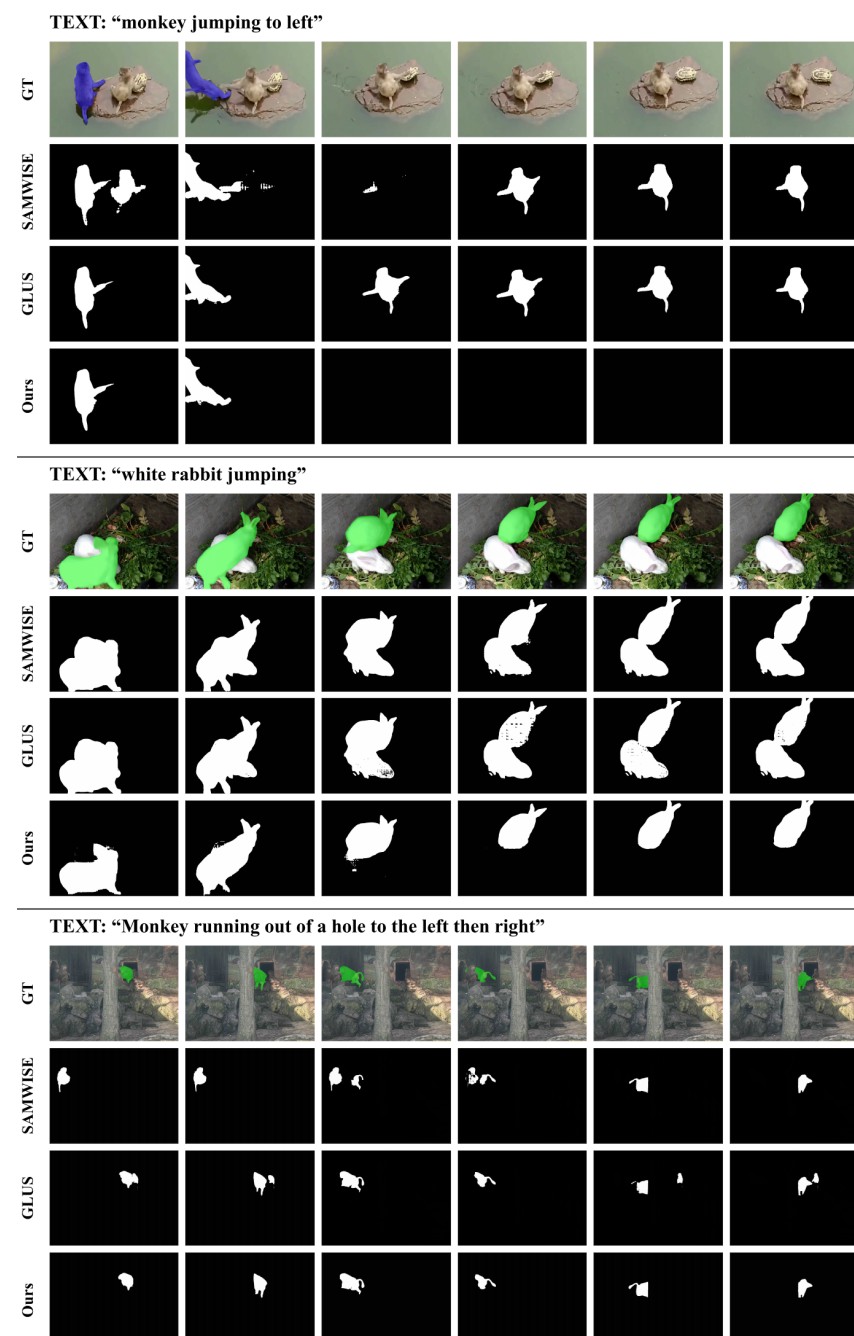

Figure 8: Comparison of qualitative results.

## A.5 LIMITATIONS

Our TGL framework introduces a moment-aware approach for RVOS, where training is performed using text-relevant frames to achieve precise video-text alignment. This stands in contrast to prior methods that sample frames randomly, often including ones unrelated to the expression, which can degrade grounding performance. By focusing on temporally aligned frames, our method learns to better associate visual content with language. However, this training strategy introduces a practical limitation: during inference, the model requires knowledge of which temporal segments are relevant to the given expression in order to accurately localize the referred object. Without such information,

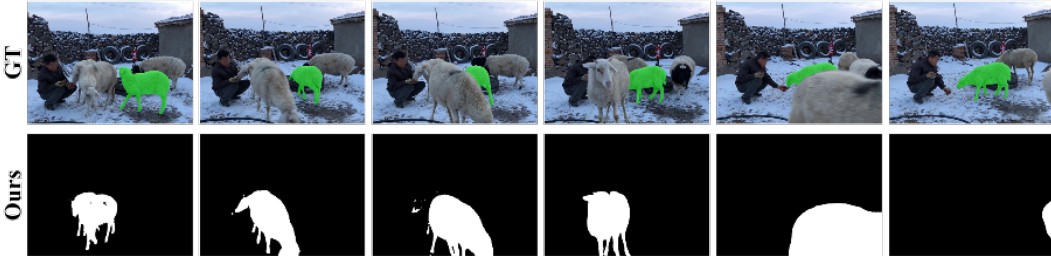

Figure 9: An example of a failure case.

its grounding capability may decline. As a result, an additional moment retrieval step—using VLMs or a moment retrieval model—is required to identify and select the most relevant segments prior to inference.

A further limitation arises when a single referring expression describes multiple temporally separated actions involving the same object, as shown in Fig. 9. Since TGL uses a single expression-level feature to attend over the entire video, it can struggle to consistently localize the object across distinct action spans, especially when one action visually dominates. This reflects a challenge in handling fine-grained temporal compositionality in multi-action expressions.

While this reliance adds overhead, our method still achieves significantly higher performance than existing approaches when accurate moments are available. This supports our core claim that moment-aware training enables more effective video-language grounding and leads to robust RVOS across diverse scenarios.

## A.6 LLM USAGE

We utilized a Large Language Model (LLM) as a writing assistant for grammatical corrections and minor improvements to enhance the clarity of this paper. Additionally, the LLM was employed to help search for missing references.

