# OpenReview forum: "Temporal Grounding as a Learning Signal for Referring Video Object Segmentation"
_ICLR.cc/2026/Conference — ICLR 2026 Conference Withdrawn Submission_

### Official Review · Reviewer_kVPy · 2025-10-26

**Soundness:** 2
**Presentation:** 2
**Contribution:** 1
**Rating:** 2
**Confidence:** 4

**Summary:**

This paper identifies a core limitation in Referring Video Object Segmentation (RVOS) – the lack of an explicit temporal learning signal during training, which leads to semantic misalignment between language expressions and video content. To address this, the authors introduce MeViS-M, a dataset augmenting the MeViS benchmark with manual annotations of the temporal spans when objects are referred to. They further propose a Temporally Grounded Learning (TGL) framework that incorporates this temporal grounding as a direct supervisory signal. TGL consists of two key strategies: Moment-guided Dual-path Propagation (MDP) for decoupling language-guided and language-agnostic processing, and Object-level Selective Supervision (OSS) for supervising only temporally-aligned objects to reduce semantic noise.

**Strengths:**

1. The paper is well-written and clearly structured, making the proposed methodology and contributions easy to understand.

2. The ablation studies validate the effectiveness of the individual components (MDP and OSS) within the proposed TGL framework.

**Weaknesses:**

1. The novelty of the proposed method appears somewhat limited. The most significant contribution seems to be the manually annotated MeViS-M dataset. While valuable, this approach is labor-intensive and may not be easily scalable. Furthermore, the testing strategy of directly using VLMs to select top-k frames is relatively straightforward and may lack sophistication.

2. Table 1 would be strengthened by including comparisons with other methods (such as SAMWISE and GLUS) under the setting of 'using VLMs to extract keyframes from ground-truth moments', which would provide a more comprehensive and fair assessment of the proposed approach's advantages.

3. The paper lacks an analysis of the computational cost and inference efficiency of the proposed framework.

**Questions:**

Please refer to the weaknesses.

---

### Official Review · Reviewer_oBhe · 2025-10-30

**Soundness:** 3
**Presentation:** 3
**Contribution:** 3
**Rating:** 4
**Confidence:** 3

**Summary:**

This paper presents a novel framework for Referring Video Object Segmentation (RVOS), termed Temporally Grounded Learning (TGL), along with a newly annotated dataset, MeViS-M, designed to provide explicit temporal signals that are typically absent during training. The authors argue that existing approaches supervise all visible objects indiscriminately, leading to semantic inconsistencies across frames. To address this issue, TGL introduces two key components: Moment-guided Dual-path Propagation (MDP), which enhances cross-modal alignment, and Object-level Selective Supervision (OSS), which refines supervision signals at the object level. Extensive experiments show that TGL achieves state-of-the-art performance on the MeViS benchmark and exhibits strong generalization to Ref-YouTube-VOS and Ref-DAVIS.

**Strengths:**

- The paper is clearly written and well-structured; the figures and tables aid comprehension, and the methodology is logically sound and presented.
- The proposed temporal grounding strategy effectively achieves leading performance on the newly introduced MEVIS-M benchmark.

**Weaknesses:**

- While the authors argue that using purely visual features (FSAM) in M- can mitigate semantic contamination, textual features in M+ are still required for memory queries during inference. Does this inconsistency across feature spaces induce the accumulation of cross-modal propagation errors?
- OSS assumes that each target object in a video corresponds to a specific time interval described by language. However, in real-world scenarios, linguistic expressions are often ambiguous, polysemous, or span multiple time periods. The forced annotations in MeViS-M may introduce “pseudo-precise” supervision signals, potentially undermining the model’s generalization ability.

**Questions:**

- It would be helpful if the authors could provide benchmark results on more general RVOS datasets, such as Ref-YouTube-VOS or Ref-DAVIS17, to evaluate the overall performance of the two modules proposed in the paper.
- When annotating MeViS-M, how was semantic relevance between a frame and its corresponding language description determined? Was there a unified annotation guideline or standard used for this process?

---

### Official Review · Reviewer_p8A9 · 2025-10-30

**Soundness:** 4
**Presentation:** 4
**Contribution:** 4
**Rating:** 2
**Confidence:** 4

**Summary:**

This work introduces TGL, a training framework for Referring Video Object Segmentation that leverages temporal moment annotations. The main contributions include the MeViS-M dataset with object-level temporal labels and two training strategies (Moment-guided Dual-path Propagation and Object-level Selective Supervision). While the dataset contribution is valuable and the technical approach is reasonable, concerns about practical applicability and experimental design limit the overall impact.

**Strengths:**

* **Valuable Dataset Contribution**: MeViS-M provides carefully curated temporal annotations with corrections to the original MeViS (fixing ID errors, adding missing objects, correcting masks as shown in Figure 4). This refined benchmark will benefit the research community.

* **Well-Articulated Problem**: The paper clearly identifies a fundamental issue—training models to segment "jumping cats" while supervising frames where cats sit still creates contradictory learning signals. This motivation is effectively communicated through Figure 1.

* **Sound Technical Design**: The dual-path propagation mechanism logically addresses feature inconsistency by using text-conditioned features (F_Adp) for relevant moments and language-agnostic features (F_SAM) for irrelevant ones. Section 4.2 provides good intuition for this design choice.

* **Thorough Experimental Analysis**: The Oracle vs. Regular comparison effectively demonstrates the method's potential and limitations. Ablation studies (Tables 2-3) systematically validate each component's contribution (+4.0 points overall improvement).

**Weaknesses:**

* **Unfair Experimental Comparisons**: All baseline methods were trained on original MeViS while TGL uses MeViS-M with corrected annotations and temporal labels. This confounds two factors: (1) cleaner training data and (2) novel methodology. To fairly assess the contribution of MDP and OSS, baselines (especially SAMWISE) should also be trained on MeViS-M with moment-aware sampling. Additionally, Table 1 shows SAMWISE with VLMs (†) performs worse than without (48.9 vs 49.5), contradicting the premise that better temporal information helps. This needs explanation.

* **Limited Scope and Generalization**: The framework is tightly coupled with SAM2 architecture. It's unclear whether MDP and OSS would transfer to other RVOS methods (ReferFormer, MTTR, DsHmp) or different foundation models. The zero-shot results (Table 5) on Ref-YouTube-VOS and Ref-DAVIS are promising but these benchmarks are simpler than MeViS. Analysis of when temporal grounding helps versus when it's unnecessary is missing.

**Questions:**

1. **Could you provide controlled comparisons** where baseline methods (particularly SAMWISE) are trained on MeViS-M with moment-aware sampling but without MDP/OSS? This would isolate the contribution of your architectural innovations from the dataset improvement.

2. **How are motion-centric embeddings (E_M) extracted?** The paper mentions "verb-related tokens" but provides no implementation details. Is this POS tagging, learned attention, or manual annotation?

3. **What explains the SAMWISE† performance drop?** Why does adding VLM-based keyframe selection hurt SAMWISE's performance?

---

### Official Review · Reviewer_ppSS · 2025-10-31

**Soundness:** 3
**Presentation:** 3
**Contribution:** 2
**Rating:** 4
**Confidence:** 5

**Summary:**

The paper tackles referring video object segmentation (RVOS) task. The authors identify the core problem as the absence of explicit temporal learning signals at training phase. Therefore, the paper introduce a new dataset MeViS-M, by manually annotating temporal spans of each referred object. To leverage temporal signal, the authors propose temporally grounded learning (TGL), which contains moment-guided dual-path propagation and object-level selective supervision. Extensive experiments demonstrate that our TGL framework effectively leverages temporal signal to establish a new state-of-the-art on the challenging MeViS benchmark.

**Strengths:**

1.	The authors introduce a new dataset MeViS-M with additional temporal annotations.
2.	The proposed TGL framework achieves state-of-the-art performance on the challenging MeViS benchmark.

**Weaknesses:**

1.	The paper presents a new approach TGL. However, it mainly introduces additional supervisory signals into the dataset. The MDP and OSS modules in TGL seem to perform selective memory computation or mask filtering based on these auxiliary signals, which may somewhat limit the overall novelty.

2.	The proposed method appears to rely mainly on the frame-level annotations provided by the temporal grounding (i.e., whether the target appears in the current frame), which may be somewhat similar to pre-defined key frame extraction and it appears that the model does not explicitly model temporal relationships.

3.	The manually annotated temporal grounding spans effectively enhance the model’s alignment between videos and referring expressions. However, in practical scenarios without temporal annotations, relying on additional dependencies (e.g., VLMs or grounding models) may introduce noise and higher computational overhead, which could potentially limit its practical value. From this point, the comparison is not fair in the table 1 and table 2.

4.	TGL shows stronger zero-shot capability compared with other methods, though a performance gap still exists relative to fully supervised approaches. Furthermore, as other datasets lack temporal annotations, it remains uncertain whether the proposed method can maintain consistent performance across different domains.

**Questions:**

1.	What are the detailed differences between the "Regular" and "Oracle" setups in the experiments? The paper indicates that “Oracle” directly uses the temporal annotations from MeViS-M at inference time. How do other competing methods utilize this information? Additionally, do other methods also leverage such temporal annotations during training?

---

### Official Review · Reviewer_184R · 2025-11-01

**Soundness:** 3
**Presentation:** 2
**Contribution:** 1
**Rating:** 2
**Confidence:** 5

**Summary:**

MeViS-M tackles the semantic misalignment in Referring Video Object Segmentation caused by the lack of explicit temporal supervision. It introduces 1) Temporally Grounded Learning with Moment-guided Dual-path Propagation and 2) Object-level Selective Supervision to leverage annotated temporal spans.

**Strengths:**

- The proposed approach outperforms previous approaches on multiple datasets at multiple metrics and settings.

**Weaknesses:**

- Architecture Novelty: The novelty looks limited with focus more on filtering the dataset than developing/modifying the architecture.
    - Section 4.2: Design Motivation: For irrelevant frames usage of direct raw predictions from SAM - wouldn’t SAM output masks for all the objects present in the scene? How would the approach filter the irrelevant masks?
    - Section 4.3: Motion-Guided Propagation: Why are M- frames being passed at train time or used if not utilized at all? I didn’t get this part. I understand at inference time M- is needed because original Mevis has it, but why is it needed/used during train time? Another query is how would M- be utilized to select and propagate masks for M-? If let’s say an example from fig 1 M+ is trained and supposed to ground only that single instance of cat jumping specifically - how would it ground other instances if it is present in original Mevis?
    - Section 4.4: Object-Level Selective Supervision:  The statement that RVOS methods would train on irrelevant objects - I don’t think it can be correct. Why would model train on let’s say cow if the query is saying cat is jumping and there’s cat and cow both present in the scene? Wouldn’t that defeat the whole purpose of the task - referral video object segmentation being referral?

 - Section 3: MeViS-M Dataset
    - What percentage of samples contain in terms of amount of average frames per video and total %age of frames which is misaligned in MeVis? How many of them are for a single object category and how many are for multiple object categories? Have authors manually annotated spatio-temporally - the location of each specific object which is actually performing the action being mentioned in the text query - spatially and temporally?

- Results and Ablations
    - Previous approaches: Are they trained on this new MeVis-M dataset? The numbers seem to be the same from original works (accumulated in Table 3 [1]). If not, then it makes an unfair comparison against previous approaches.
    - Table 5: Are the previous approaches trained on Mevis and then zero=shot evaluated on Ref-Youtube-VOS and Ref-DAVIS?


[1] Ding, H., Tang, S., He, S., Liu, C., Wu, Z., & Jiang, Y. G. (2025). Multimodal referring segmentation: A survey. arXiv preprint arXiv:2508.00265.

**Questions:**

Please see the weakness section.

---

### Note · Authors · 2025-11-12

I have read and agree with the venue's withdrawal policy on behalf of myself and my co-authors.